# Exploring Gut Microbiome in Predicting the Efficacy of Immunotherapy in Non-Small Cell Lung Cancer

**DOI:** 10.3390/cancers14215401

**Published:** 2022-11-02

**Authors:** Ben Liu, Justin Chau, Qun Dai, Cuncong Zhong, Jun Zhang

**Affiliations:** 1Department of Electrical Engineering and Computer Science, University of Kansas, Lawrence, KS 66045, USA; 2Division of Hematology, Oncology, and Blood & Marrow Transplantation, University of Iowa Hospitals and Clinics, Iowa City, IA 52242, USA; 3Division of Medical Oncology, Department of Internal Medicine, University of Kansas Medical Center, Kansas City, KS 66160, USA; 4Bioengineering Program, School of Engineering, University of Kansas, Lawrence, KS 66045, USA; 5Center for Computational Biology, University of Kansa, Lawrence, KS 66045, USA; 6Department of Cancer Biology, University of Kansas Medical Center, Kansas City, KS 66160, USA

**Keywords:** immunotherapy, non-small cell lung cancer, gut microbiome, prediction model, machine learning

## Abstract

**Simple Summary:**

Despite the emerging success of immunotherapy in non-small-cell lung cancer (NSCLC), it remains clinically important to better identify patients who are likely to respond to treatment, especially considering the existence of immune-related adverse events (irAEs). In recent years, the gut microbiome has been correlated with treatment response, but no predictive models relating the two have been developed. In this study, we used random forest and neural networks to predict the progression-free survival of NSCLC patients treated with immunotherapy. Our results showed that a functional profile of the human gut microbiome outperformed the taxonomical profile across different studies, which can be utilized to establish a model with good predictive value in lung cancer immunotherapy.

**Abstract:**

We performed various analyses on the taxonomic and functional features of the gut microbiome from NSCLC patients treated with immunotherapy to establish a model that may predict whether a patient will benefit from immunotherapy. We collected 65 published whole metagenome shotgun sequencing samples along with 14 samples from our previous study. We systematically studied the taxonomical characteristics of the dataset and used both the random forest (RF) and the multilayer perceptron (MLP) neural network models to predict patients with progression-free survival (PFS) above 6 months versus those below 3 months. Our results showed that the RF classifier achieved the highest F-score (85.2%) and the area under the receiver operating characteristic curve (AUC) (95%) using the protein families (Pfam) profile, and the MLP neural network classifier achieved a 99.9% F-score and 100% AUC using the same Pfam profile. When applying the model trained in the Pfam profile directly to predict the treatment response, we found that both trained RF and MLP classifiers significantly outperformed the stochastic predictor in F-score. Our results suggested that such a predictive model based on functional (e.g., Pfam) rather than taxonomic profile might be clinically useful to predict whether an NSCLC patient will benefit from immunotherapy, as both the F-score and AUC of functional profile outperform that of taxonomic profile. In addition, our model suggested that interactive biological processes such as methanogenesis, one-carbon, and amino acid metabolism might be important in regulating the immunotherapy response that warrants further investigation.

## 1. Introduction

Lung cancer remains the leading cause of cancer-related death in the US and worldwide [1,2]. With a better understanding of immune checkpoints in tumor control, immunotherapy using immune checkpoint inhibitors (ICIs) has revolutionized our treatment in various types of cancer, including lung cancer [3,4,5]. Pembrolizumab, for example, binds to and impairs the lymphocyte PD-1 receptor’s ability to interact with PD-L1 on cancer cells and thus allows the enhancement of antitumor immune response via suppression of the co-inhibitory PD-1/PD-L1 pathway [6], which has resulted in a significantly better response and survival in certain patients with advanced/metastatic NSCLC [7,8,9]. However, not all patients benefit from ICIs, and some even develop non-negligible irAEs [10,11], such as potentially lethal pneumonitis. Given this, it is desirable to develop predictive models that better identify patient populations in which treatment benefits outweigh risks.

Several studies, including ours, have associated the gut microbiome with the host’s immune system and immunotherapy response and irAEs [12,13,14,15,16]. For example, *Bacteroides thetaiotaomicron* and *Bacteroides fragilis* were reported to be positively associated with the efficacy of CTLA-4 blockade [17]. Enrichment of *Bifidobacterium* was reported to be associated with a lower incidence of irAEs in lung cancer patients receiving ICIs [18]. Oral intake of *Bifidobacterium* combined with anti–PD-L1 antibody therapy showed significant improvement in melanoma control in mouse models [19], and *Bifidobacterium* was also reported to suppress metastasis of lung cancer in mouse models [20]. *Akkermansia muciniphila* was found enriched in NSCLC patients who responded to PD-1–based immunotherapy [21]. However, several key questions remain, the external validity of a study’s taxonomic analysis of the gut microbiome, especially considering the existence of various modulating factors [22]; can functional analysis provide a better signal considering the inherent functional redundancy of microbiota, and if so, whether such a signal can have predictive value? Up to now, there is no published prediction model using the gut microbiome to predict the efficacy of immunotherapy in NSCLC. The prediction model proposed for melanoma did not show satisfactory performance, with a 71% F1 score on the testing set [23]. 

The heterogeneity of human gut microbiome taxonomical composition challenges its predictive use [23]. For example, Limeta, et al. showed that the weighted UniFrac distances were smaller in patients’ gut microbiomes when grouping by study than by response [23]. Similar findings were reported in healthy humans by the Human Microbiome Project [24]. However, despite the dissimilarity of taxonomical compositions, Huttenhower et al. did show that most metabolic pathways were commonly shared across different human subjects [24]. 

The possibility of predicting rather than merely associating the efficacy of immunotherapy in NSCLC patients using the taxonomic and/or functional gut microbiome as a biomarker collectively intrigued us. To answer this, in the current study, we performed analyses on the taxonomic and functional features of the gut microbiome between long PFS (above 6 months) and short PFS (below 3 months) and trained a random forest (RF) classifier and a multilayer perceptron (MLP) neural network classifier to predict if a patient will develop long vs. short PFS. We stratified PFS in this way to separate patients who clearly benefit from immunotherapy from those who do not benefit in an effort to enrich potential signals. Our study showed that the RF classifier achieved an 85.2% F-score and a 95% area under the receiver operating characteristic curve (AUC) using the Pfam profile, and the MLP classifier achieved a 99.9% F-score and 100% AUC using the Pfam profile. Figure 1 describes our study schema.

## 2. Materials and Methods

### 2.1. Data Set and Metadata Collection

We utilized two datasets (DS1 and DS2) in this study. The detail of DS1 is shown below in patients and samples. For DS2, we began by performing a literature search on the SRA (Sequence Read Archive) database in the NCBI (National Center for Biotechnology Information) using the search term “(NSCLC gut) AND bioproject_sra[filter] NOT bioproject_gap[filter]” with a cutoff date of 14 December 2021, to retrieve all publications related to NSCLC immunotherapy with published metagenomic data of gut microbiome. This resulted in 5 records. We found that 2 out of the 5 records contained whole metagenome shotgun sequencing (WGS) data, and only one [21] has metadata associated with the WGS data, which was referred to as DS2 in this paper. Broadening the search criteria to “lung cancer” resulted in 16 records, but no further usable WGS data containing metadata was found.

### 2.2. Patients and Samples

In total, 14 NSCLC patients who received immunotherapy and have known PFS were selected from our previous study [18] (referred to as DS1, which can be accessed from PRJNA866654). Their pretreatment baseline fecal samples were collected and extracted DNA sequenced on Illumina HiSeq for 2 × 150 pb; NexteraXT preparation was used by COSMOSID^®^. DS2 contained 65 additional pretreatment samples. For the PFS study, 6 patients with long PFS and 3 patients with short PFS were included in DS1, and 7 samples with long PFS and 34 samples with short PFS were included in DS2. We omitted patients with PFS between 3 and 6 months to further contrast the gut microbiome of patients with long vs. short PFS. We later applied our trained models to predict treatment response. We grouped patients with a complete and partial response as responders (R), whereas those with stable disease, progression, and death as non-responder (NR). With this design, all 79 patients (14 from DS1 plus 65 from DS2) were included for analysis, with 8 R and 6 NR in DS1 and 12 R and 53 NR in DS2. RECIST 1.1 criteria [25] were used to assess the treatment response.

### 2.3. Quality Control, Annotation, and Differential Study

All raw metagenomic data were quality trimmed using Trimmomatic [26] (v0.38) using a user-specific adapter list with default parameters (Appendix A). The potential human genome sequences were removed using BWA [27] (0.7.16a-r1185-dirty) with default parameters (the percentage of reads removed was reported in Appendix A). MetaPhlAn [28] (v3.0.6) was used to annotate the taxonomic composition against its own database mpa_v30_CHOCOPhlAn_201901 (containing marker genes from ~99,500 bacterial and ~500 eukaryotic genomes). UProC [29] (v1.2.0) was used to estimate the abundances of the Pfam families [30] (28.0) and KEGG Orthology (the March 2014 release). MicrobiomeAnalyst [31] was used to compute alpha-diversity and beta-diversity, and metagenomeSeq [32] was used to identify differentially abundant taxa. Significant differential-abundance Pfam and KEGG Orthology families were determined by DESeq2 [33] (v1.22.2). Hierarchical clustering was then performed using clustermap in seaborn [34] (v0.11.0), with the Z-scores obtained from normalized family-level RPKM (Reads Per Kilobase Million). 

### 2.4. Prediction

The RF and MLP classifiers from scikit-learn [35] (v0.24.2) were used to classify the patients into long/short PFS groups and between R/NR groups. We selected RF for its interpretability and MLP for its regression power [36]. Training-testing datasets for PFS classification were constructed such that the training and testing datasets contained 35 and 15 samples, respectively, and the testing dataset contained 4 samples with long PFS. Gini importance [37] was adopted to reduce the input dimension and avoid model overfitting. The peak AUC performances of taxa-, KEGG Orthology-, Pfam-, and the combined information-based models were achieved at 7, 60, 38, and 18 selected features, respectively (Appendix A). An early stop was enabled for MLP to prevent overfitting to the training set; all the other parameters were left as default.

We define true positives (TP) as the number of patients who were predicted and indeed had PFS > 6 months, and the true negatives (TN) as those predicted and indeed had PFS < 3 months. Correspondingly, the false positives (FP) corresponded to the patients predicted with long but had short PFS, and vice versa for the false negatives (FN). The performances of our models were measured by sensitivity, precision, F-score, and accuracy. The receiver operating characteristic (ROC) curves were generated from data collected from 100 repeated experiments. When calculating the AUC score, they were extrapolated to the extreme points corresponding to the highest sensitivity and precision. 

Our Pfam-based RF and MLP models trained on 50 PFS-labeled patients were further used for R/NR classification. The test set contained 79 samples with R/NR group labels. A null model (random guess) was repeated 1000 times to simulate the background distribution. Detailed scripts and comments can be found in the Appendix A.

## 3. Results

### 3.1. Taxonomic Differences between the Long and Short PFS Gut Microbiomes

All of the raw sequencing data were re-processed and re-analyzed using the pipeline described in Figure 1 and Methods. After combining the two datasets, examining the taxonomic distribution at the phylum level did not show a significant difference between long vs. short PFS groups. Comparisons of alpha-diversity and beta-diversity at the genus level also showed no significant difference (Figure 2A,B). 

Given that no significant difference was observed at the global level using merged datasets (Figure 2A,B), Appendix A), we individually compared the taxonomic profiles between PFS groups in DS1 and DS2. We used taxa with the largest differentials in abundance between the groups for hierarchical clustering (Figure 2C,D), also in Appendix A). In total, 23 taxa of interest were found in DS1 (*p*-value ≤ 0.05, Appendix A), and 24 were found in DS2 (*p*-value ≤ 0.05, Appendix A). Two long PFS samples (JZLC_19 and JZLC_37) demonstrated significantly different taxonomic profiles than the other samples within the same group (Appendix A) and thus were excluded from further analysis. The taxa with *p*-value ≤ 0.05 successfully classified PFS in DS1 but failed in DS2 (Figure 2C,D), suggesting taxonomic profile alone might not be enough to cluster/predict long vs. short PFS. 

### 3.2. Functional Differences between Long and Short PFS Gut Microbiomes

We then investigated the use of the Pfam and KEGG Orthology protein families for long/short PFS classification. In total, 8516 protein families from Pfam were detected in DS1 and DS2, whereas 10,806 were detected from KEGG Orthology (Appendix A). Moreover, 171 and 163 protein families showed significantly different abundance (*p*-value ≤ 0.05, Appendix A) in DS1 from the Pfam and KEGG Orthology, respectively; while 213 and 239 of them showed significantly different abundances (*p*-value ≤ 0.05, Appendix A) in DS2 from Pfam and KEGG Orthology, respectively. The top 50 protein families of Pfam or KEGG Orthology clustered most of the long vs. short PFS patients (Figure 3, also in Appendix A), Appendix A), with only two misclassifications (Figure 3D).

### 3.3. PFS Prediction Using Taxonomic and Functional Information

Having identified that the functional profile of the gut microbiome can better segregate long vs. short PFS, we next investigated whether it could have better predictive power than taxonomic information. Figure 4A–D shows the distribution of RF prediction scores (ranging from 0 to 1) of 15 testing samples (recall the training-testing splitting above) in 100 experiments using the taxonomic, KEGG Orthology, Pfam, and combined profiles, respectively (the true positive cases were marked in orange). The functional profiles (KEGG Orthology and Pfam) clearly outperformed the taxonomic profile. By using different cutoffs of prediction scores, the receiver operating characteristic (ROC) curve of average performances was shown in the left panel of Figure 4E, from which we can see that the Pfam profile achieved the best AUC (95%) and F1 score (85.2%, the harmonic mean of the precision and sensitivity (see right panel of Figure 4E). This is reflected in the distribution pattern of positive cases in Figure 4A–D, showing a better separation of positive vs. negative predictive events for PFS. Compared to the KEGG Orthology profile (Figure 4B), the prediction scores of positive cases using the Pfam profile (orange dots in Figure 4C) were higher, and the majority of the negative cases (grey dots) were assigned with lower prediction scores (accumulated to the left side). This feature of the Pfam profile might be clinically useful: a high score (more than 0.5) implies an NSCLC patient is very likely to have long PFS on immunotherapy, whereas a low score implies otherwise. However, the existence of such a lower-score cutoff needs to be verified using more samples from prospective studies. Figure 4E shows the value of performance matrices using the default cutoff of prediction score (0.5), and the highest values were in bold. The Pfam profile achieved the best F1 score (85.2%) and best accuracy (92.8%).

To maximize the performance, we applied MLP to predict PFS. The ROC curve and performance matrices were computed using identical setups as RF. The results are shown in Figure 4G) (also Appendix A), from which we observed that the Pfam, KEGG Orthology, and combination profiles achieved nearly perfect predictions in the range of 97% to 100% for the AUC and 98.4% to 99.9% for the F1 score. The predictive power of the taxonomic profile was again considered inferior.

### 3.4. Treatment Response Prediction Using Pfam-PFS Model

As quite often (although not always) in the clinical setting, we observe that treatment response correlates with survival benefits (e.g., longer PFS). Thus, we further studied the predictive power of the Pfam-PFS feature set in treatment response. To do that, we directly applied the RF and MLP classifiers, trained by the 50 samples with PFS labels, to the 79 samples with R/NR labels. We benchmarked the trained classifiers against a null (random) predictor and simulated the performance of the stochastic predictor 1000 times to approximate the background distribution. The performances of the Pfam-based RF and MLP classifiers were shown as the red crosses in Figure 4F,H, respectively. The trained RF and MLP classifiers statistically significantly outperformed the stochastic predictor in F-score (with a *p*-value less than 1 × 10^−6^). It should be noted that none of the responder labels was known by the classifiers during the training process; thus, Figure 4F,H suggested the potential to develop a single model that predicts both PFS and treatment response. 

### 3.5. Biological Processes with Potential Impacts on NSCLC Immunotherapy Response

Noticing that the functional profiles can better predict the immunotherapy response, we explored its relevant biological processes. Table 1 lists all statistically significant biological processes from the 38 Pfam protein families used for prediction (see Appendix A for 38 Pfam protein families and full list of biological processes).

#### 3.5.1. Methanogenesis and One-Carbon Metabolic Process

Out of 14,107 protein families, we found that 3 out of 14 methanogenesis-related protein families [30] were enriched in patients with long PFS (*p*-value 2.11 × 10^−5^) (PF02249, PF02240, and PF02505, all related to methyl-coenzyme M reductase: MCR). Methanogenesis is the formation of methane by microbes known as methanogens, which are primarily belonging to the *Archaea* domain [38], and MCR is the key enzyme of this biological process [39]. Methanogenic *Archaea* inhabit mammals’ gastrointestinal (GI) tract and have syntrophic interactions with other microorganisms within the microbial community [40]. Some of them, for example, *Methanobrevibacter smithii*, can be recognized by the human innate immune system and activate dendritic cells, therefore contributing to the activation of the adaptive immune response [41]. In addition, methanogenic *Archaea* can be functional associates of the fermentative digestion of dietary fibers, favoring the production of beneficial short-chain fatty acids [42] that is associated with good immunotherapy response [43]. Consistent with this, several studies have positively correlated methanogenic microbiota with cancer immunotherapy [44,45]. In fact, our taxonomic analysis also showed that *Methanobrevibacter smithii* is one of the top enriched microbiota in patients with long PFS. 

We also found that 1 out of 4 one-carbon metabolism families was significantly enriched (*p*-value of 0.01719) (PF02741, annotates the proximal lobe of formylmethanofuran--tetrahydromethanopterin formyltransferase: FTR). Considering that methanogenesis is a process converting bacterial metabolic products (e.g., CO_2_, formate, etc.) to methane, it is not surprising to see the importance of the one-carbon metabolic process as it is instrumental in reducing CO_2_ (the most oxidized one-carbon compound) to methane (the most reduced form of a one-carbon compound), which is accompanied with electrons derived from the oxidation of either H_2_ or formate [46]. In fact, FTR participates in both methanogenesis and folate biosynthesis. Interestingly, one-carbon metabolism has recently been shown to play an essential role in T-cell function. For example, adding products of one-carbon metabolism (such as formate and glycine) was found capable of enhancing the activation of aged naïve T cells [47]. The deficiency of folate, which supports the one-carbon metabolism, has been shown to substantially reduce CD8+ T cell (cytotoxic T cell, CTL) proliferation [48]. Furthermore, methyl-B12 can promote both the number and activity of CD8+ T cells [49,50]. Of note, vitamin B12 is a cofactor for methionine synthase and contributes to the one-carbon metabolism [51].

#### 3.5.2. Amino Acid Biosynthetic and Metabolic Processes 

We also found that three processes were statistically significant in regard to amino acids. The branched-chain amino acid (BCAA) biosynthetic process (PF01450, annotates the catalytic domain of acetohydroxy acid isomeroreductase: AHIR) and cellular amino acid biosynthetic process (PF07991, annotates the NADPH-binding domain of AHIR) are two processes relevant to AHIR (also known as ketol-acid reductoisomerase, KARI). AHIR not only participates in the formation of BCAAs such as isoleucine, leucine, and valine, but it also catalyzes the reversible transformation of NADP+ and NADPH [52]. Of note, NADPH is reported to be an additional product of one-carbon metabolism [53]. The cellular amino acid metabolic process is related to Pfam PF00742, which annotates homoserine dehydrogenase that catalyzes the third step in the aspartate pathway [54,55]. The aspartate pathway produces essential amino acids threonine, methionine, lysine, and isoleucine; the cofactor S-adenosylmethionine; and the cell wall component diaminopimelate. The third step of the aspartate pathway is the NAD(P)-dependent reduction of aspartate beta-semialdehyde into homoserine. Homoserine is an intermediate in the biosynthesis of threonine, isoleucine, and methionine.

Amino acids are found important to support immunity by providing energy or biomass to support the proliferation of immune cells and via modulation of key metabolic pathways that instruct immune cell function [56]. For example, BCAAs such as leucine, isoleucine, and valine can provide acetyl-CoA and succinyl-CoA that enter the TCA cycle [57], and supplementation of BCAA could enhance CD8+ T cell activity [58]. Amino acids can also be used to make antioxidants such as glutathione to maintain redox balance and provide methyl and acetyl groups to epigenetically regulate gene expression patterns in immune cells [56]. 

Interestingly, methanogens can generate serine during methanogenesis and synthesize lysine [59], and one-carbon metabolism directly modulates the levels of three amino acids: methionine, serine, and glycine [53], and connects the TCA cycle via NADH [60]. All these suggest a close interaction among these biological processes. Since many metabolites/metabolic intermediates can reach host cells (including immune cells) from gut microbiota, and several metabolic pathways such as one-carbon metabolism span all kingdoms [61], these biological processes could also prime host immune cells to respond better to immunotherapy (Figure 5).

#### 3.5.3. Plasmid Maintenance

This significantly enriched biological process in patients with long PFS is due to Pfam PF05732, which annotates firmicute plasmid replication protein (RepL). Firmicutes were reported to be positively associated with better immunotherapy response [18,22]. In addition, lower plasmid diversity is associated with gut dysbioses such as inflammatory bowel disease (IBD), and higher plasmid diversity is associated with higher alpha diversity, which also correlates with a healthier condition and, in general, better response to immunotherapy [62]. 

## 4. Discussion

Although previous studies, including ours [18,21,22,63] have demonstrated the correlation of the gut microbiome with immunotherapy response, this is arguably the first study showing that the gut microbiome can be used to predict treatment response in lung cancer immunotherapy. We have illustrated its potential to predict long vs. short PFS and treatment response in NSCLC patients receiving ICIs. This study also showed that the functional profile, particularly the Pfam profile, outperformed the taxonomic profile by 3.9% in F-score and 11% in AUC using RF and by 10.4% in F-score and 14% in AUC using MLP. This can be explained by the fact that the Pfam profile is more granular than the taxonomic profile: Pfam annotations used more information contained in the raw data. 

We also noticed that using Pfam, several biological processes such as methanogenesis, amino acid biosynthesis/metabolic process, and one-carbon metabolism were significantly enriched in patients who benefited from immunotherapy. This finding is supported by previous studies which demonstrated the importance of amino acids, folate, and cobalamin in CTLs [47,48,49,50,51], a key player in immunotherapy using ICIs, as well as shared biological processes across kingdoms such as one-carbon metabolism [61]. Though exciting, such a finding will need to be validated in future larger datasets and mechanistically using mouse models and relevant in vitro studies.

The prediction of PFS and response using the Pfam profile (Figure 4) could have significant clinical value. Note that the precision scores of the long PFS and responder were nearly 100%, meaning that patients who were predicted to benefit from immunotherapy indeed did so. On the other hand, by lowering the prediction threshold, for example, 0.2 in Figure 4C, all patients who benefitted from the therapy were accurately predicted. These suggest the possibility of setting another lower bound to predict patients who are less likely to benefit from the therapy, allowing other therapeutic approaches to be considered in advance. 

We must admit that the sample size limits our findings. With a larger sample size, the prediction models (both RF and MLP) could learn better to generalize to a larger population. We are actively enrolling patients through clinical trials (e.g., NCT04636775) with the plan to further train and validate our predictive model through continuous integration of new data. Since gut microbiome can be affected by diet and various lifestyle factors, we also plan to incorporate published data from studies performed in other geographic locations. Furthermore, since UProC [29] is not the only approach to analyze protein sequences, to minimize research method bias, we also plan to integrate other approaches, such as HUMAnN (the HMP Unified Metabolic Analysis Network) [28], in our future studies.

## 5. Conclusions

Gut microbiome may predict therapeutic benefit from immunotherapy in NSCLC patients. Its derived functional profile (e.g., Pfam) seems to have more potent predictive power than taxonomic information. The revealed biological processes, especially one-carbon metabolism, might modulate cancer immunotherapy response, which deserves validation and mechanistic investigation in future studies.

In the future, we will continue to incorporate more data to improve and validate our predictive model. Equally important, we have planned a series of mechanistic studies to understand the value of methanogenesis, one-carbon and amino acids metabolism, and archaea in modulating host immune status and response to cancer immunotherapy.

## Figures and Tables

**Figure 1 cancers-14-05401-f001:**
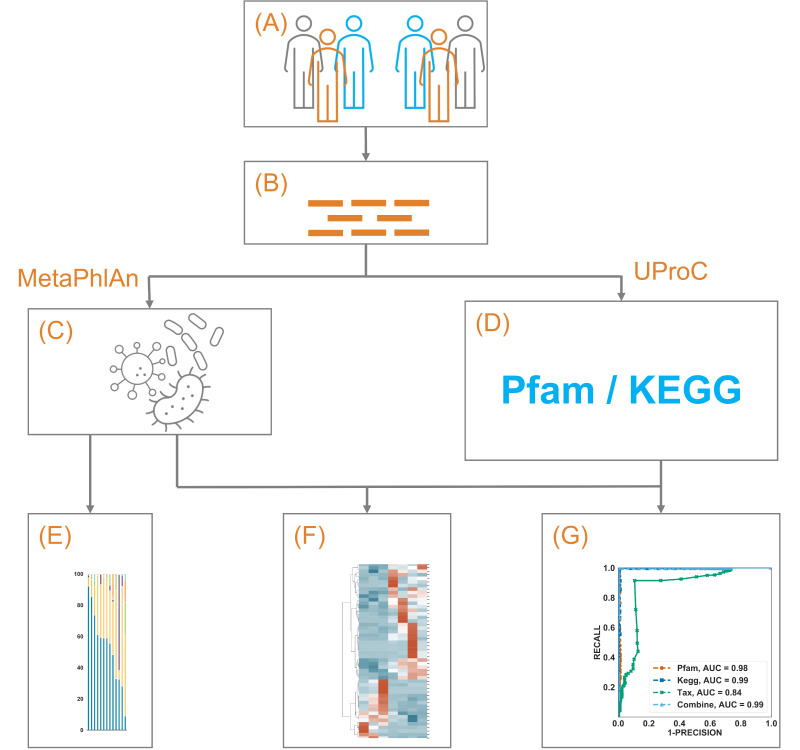
A schematic illustration of the current study. (**A**) 79 NSCLC patients from two separate studies treated with immunotherapy were included. By excluding those with PFS above 3 months but below 6 months, 9 patients (6 with PFS above 6 months) were included in dataset 1 (DS1), and 41 patients (7 with PFS above 6 months) remained in DS2. (**B**) Whole metagenome shotgun sequencing of the gut microbiome from those patients was obtained. (**C**) The microbiome taxonomic profile was constructed using MetaPhlAn and its own database. (**D**) The microbiome functional annotation profile was constructed using UproC against Pfam and KEGG Orthology. (**E**–**G**) showed the exploration, clustering, and prediction using those profiles, respectively.

**Figure 2 cancers-14-05401-f002:**
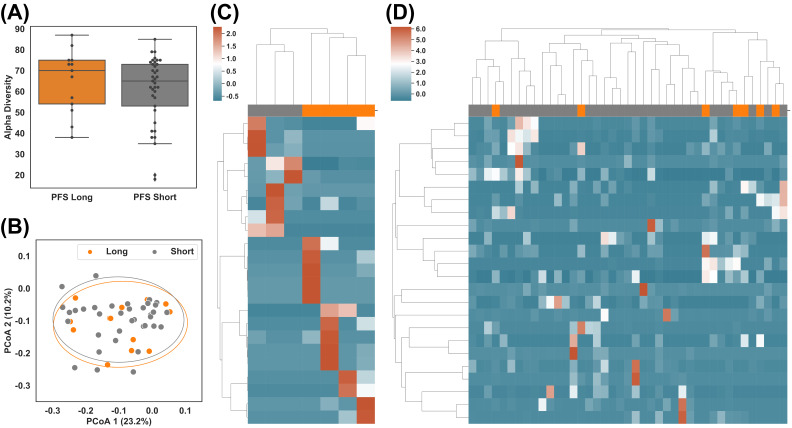
Taxonomic difference of gut microbiome between patients with long (>6 months) and short (<3 months) PFS. (**A**) Alpha diversity of subgroups after combining datasets DS1 and DS2: PFS Long (above 6 months) vs. PFS Short (below 3 months). (**B**) Beta diversity of subgroups: PFS Long vs. PFS Short. (**C**,**D**) Hierarchical clustering of z-score of abundances of the most differential taxa between PFS Long and PFS Short in DS1 and DS2, respectively.

**Figure 3 cancers-14-05401-f003:**
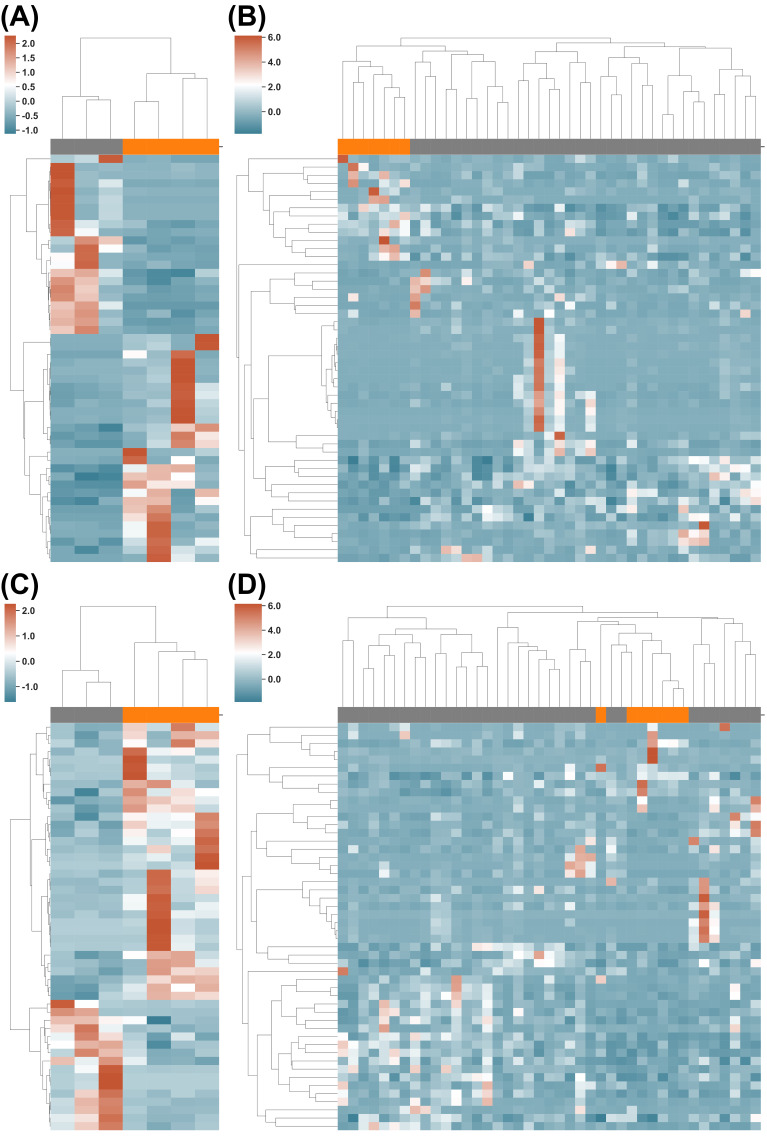
Functional differences in gut microbiome between long vs. short PFS patients. (**A**,**B**) Clustering of PFS Long (orange) and PFS Short (grey) using the KEGG Orthology profile on DS1 and DS2, respectively. (**C**,**D**) Clustering of the PFS Long and PFS Short using the Pfam profile on DS1 and DS2, respectively.

**Figure 4 cancers-14-05401-f004:**
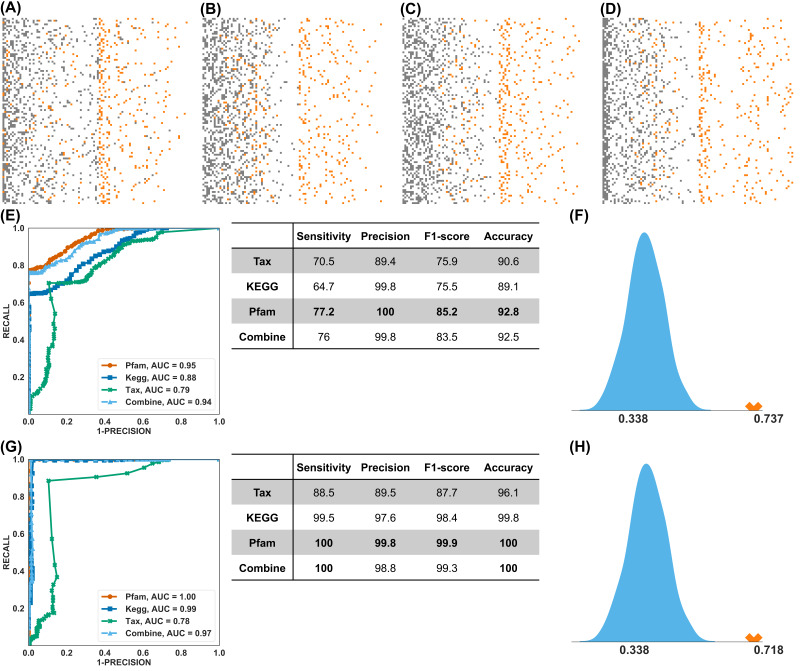
Prediction. (**A**–**D**) RF prediction score distribution using the taxonomical profile, KEGG Orthology profile, Pfam profile, and combined profiles. Each row represents one random testing experiment, and there are 100 experiments in total. Orange dots denote PFS Long samples, and grey dots PFS Short samples. (**E**) Performance of RF prediction. The left panel shows the averaged ROC curve of 4 profiles; the right panel shows the averaged prediction score using the default (0.5) prediction score cutoff. (**F**) Using the RF and Pfam feature set from the PFS study to predict responders, measured by the F1-score. The light blue curve is the exact distribution of random guessing, and the orange cross is the actual performance of the trained model. (**G**) Performance of the MLP prediction. (**H**) Using the MLP and Pfam feature set from the PFS study to predict the treatment response.

**Figure 5 cancers-14-05401-f005:**
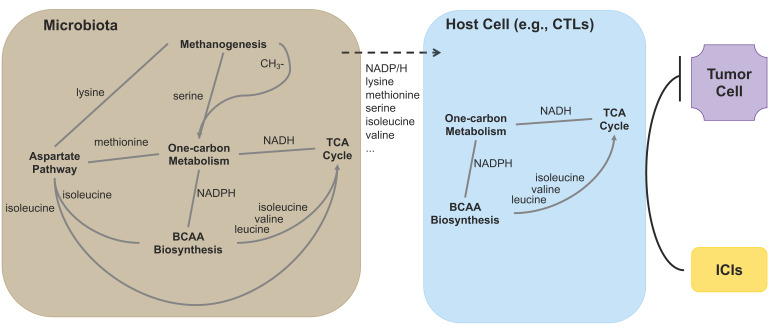
Biological processes that are enriched in patients with long PFS, their interaction, and shared components (e.g., One-carbon metabolism, BCAA biosynthesis, and TCA cycle) between gut microbiota and host cells (e.g., T cells). Methanogenesis generates serine, which enters one-carbon metabolism, and lysine, which enters the aspartate pathway. The aspartate pathway produces methionine which enters the methionine cycle of one-carbon metabolism. In addition, along with BCAA biosynthesis, it generates isoleucine that feeds the TCA cycle. BCAA biosynthesis pathway also produces valine and leucine that enter the TCA cycle. Finally, NADH/NADPH connect one-carbon metabolism, TCA cycle, and BCAA biosynthesis pathway—three important biological processes that are shared by host cells, including immune cells such as CTLs. These pathways can integrate metabolites/metabolic intermediates directly from gut microbiota, such as NADPH, serine, methionine, etc., and prime CTLs to respond better to ICIs in cancer immunotherapy.

**Table 1 cancers-14-05401-t001:** Statistically significant (*p*-value < 0.05) biological processes from the 38 Pfam protein families used for prediction.

Pfam ID	Biological Process	*p*-Value *
PF02249, PF02240, PF02505	Methanogenesis	2.11 × 10^−5^
PF01450	Branched-chain amino acid biosynthetic process	0.008611
PF07991	Cellular amino acid biosynthetic process	0.01286
PF05732	Plasmid maintenance	0.01286
PF02741	One-carbon metabolic process	0.01708
PF00742	Cellular amino acid metabolic process	0.03761

* All protein families are enriched in the long PFS group.

## Data Availability

Raw data used for analysis has been uploaded to the (NCBI) BioSample database under BioProject ID PRJNA866654, currently pending review and publication.

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
