# Peer review of "Exploring Gut Microbiome in Predicting the Efficacy of Immunotherapy in Non-Small Cell Lung Cancer"

_cancers, 2022, doi:10.3390/cancers14215401_

Round 1

Reviewer 1 Report

Authors have proposed a predictive model of immunotherapy-based efficacy in non-small cell lung cancer by exploring gut microbiome. The article is quite interesting and designed on the emerging multilayer perceptron neural network and deep learning based model.

In my opinion some query and suggestions are required before publishing this manuscript:

1.  Title of the manuscript highlighted the predictive model used for NSCLC, can we use this model for other type of lung cancer too?

2. Keywords should be limited as per the journal’s guidelines.

3. Please explain the reason: “Predictive model based on functional (e.g. Pfam) rather than taxonomic profile might be clinically useful to predict whether a NSCLC patient will benefit from immunotherapy”.

4. Introduction section: Authors can refer to the recently published article for addition of relevant content related to immunotherapy: https://doi.org/10.3390/ph15030335

5. Authors have explained the limitations of this model in terms of patient size. Please explain other limitations of this model too, if any.

6. Authors are suggested to improve the conclusion section by adding future perspectives of this study.

6. The authors are advised to recheck the whole manuscript for improving the typographical and grammatical errors carefully.

Author Response

Dear Reviewer,

Thank you for your kind comments. We have revised our manuscript per the comments, corrected some other errors, and are now re-submitting it for your consideration for publication. Below is our point-by-point response to the comments:

  1. Title of the manuscript highlighted the predictive model used for NSCLC, can we use this model for other type of lung cancer too?

Response: Good question! We do believe there are shared microbiome features associated with better response across different types of lung cancer, however, this will need to be tested when we have enough sample size of other type of lung cancer such as SCLC (small cell lung cancer).

  1. Keywords should be limited as per the journal’s guidelines.

Response: We have now reduced the number of keywords to 5 per the journal’s guideline. 

  1. Please explain the reason: “Predictive model based on functional (e.g. Pfam) rather than taxonomic profile might be clinically useful to predict whether a NSCLC patient will benefit from immunotherapy”.

Response: Although PD-L1 expression is useful in predicting immunotherapy response, it is not perfect, and there is no predictive biomarker for immune related adverse events (irAEs). People therefore started exploring the potential use of microbiome (primarily the taxonomic profile) due to its correlation with treatment response and irAEs (e.g. our previous publications: Chau J, et al. BMC Cancer 2021; Huang C, et al. Front Oncol. 2021). Since in our current study the predictive models based on functional (e.g. Pfam) are consistently better than the taxonomic profile (Figure 4), we therefore claim such models will be more clinically useful in predicting whether a NSCLC patient will benefit from immunotherapy.

  1. Introduction section: Authors can refer to the recently published article for addition of relevant content related to immunotherapy: https://doi.org/10.3390/ph15030335.

Response: We have added this article in the Introduction section.

  1. Authors have explained the limitations of this model in terms of patient size. Please explain other limitations of this model too, if any.

Response: We have expanded the discussion and added following limitations: “Since gut microbiome can be affected by diet and various lifestyle factors, we also plan to incorporate published data from studies performed in other geographic locations. Furthermore, since UProC is not the only approach to analyze protein sequences, to minimize research method bias, we also plan to integrate other approaches such as HUMAnN (the HMP Unified Metabolic Analysis Network) in our future studies.”

  1. Authors are suggested to improve the conclusion section by adding future perspectives of this study.

Response: We have improved the Conclusions section and added following future perspectives: “In the future, we will continue to incorporate more data to improve and validate our predictive model. Equally important, we have planned series of mechanistic studies to understand the value of methanogenesis, one-carbon and amino acids metabolism, as well as archaea in modulating host immune status and response to cancer immuno-therapy.”

  1. The authors are advised to recheck the whole manuscript for improving the typographical and grammatical errors carefully.

Response: We have rechecked the whole manuscript and hopefully have improved the typographical and grammatical errors significantly.

Reviewer 2 Report

Exploring gut microbiome in predicting efficacy of immunotherapy in non-small cell lung cancer

The paper had a bit of a simple confusing summary, from which one could not tell that the work here is fully computational. That being said, the report is solid. 

One very minor note on the title: Exploring gut microbiome in predicting THE efficacy of immunotherapy in non-small cell lung cancer.

Line 110 - I particularly liked that the work combined primary data from the author and data collected from a literature search (DS2). 

Generically about all figures that are Hierarchical clustering plots (e.g., Figures 2C & D). These plots must include notes and legends to help readers understand the conclusions. 

Line 306 and 310. There are multiple places where it says "Error! Reference source not found" . 

Author Response

Dear reviewer,

Thank your for your kind comments. We have revised our manuscript per the comments, corrected some other errors, and are now re-submitting it for your consideration for publication. Below is our point-by-point response to the comments:

  1. The paper had a bit of a simple confusing summary, from which one could not tell that the work here is fully computational. That being said, the report is solid.

Response: Thanks for the suggestion. We have now clarified using the following text: “In this study, we used random forest and neural networks to predict the progression-free survival of NSCLC patients treated with immunotherapy. Our results showed that a functional profile of the human gut microbiome outperformed the taxonomical profile across different studies, which can be utilized to establish a model with good predictive value in lung cancer immunotherapy.”

  1. One very minor note on the title: Exploring gut microbiome in predicting THE efficacy of immunotherapy in non-small cell lung cancer.

Response: This has been corrected. Thank you!

  1. Line 110 - I particularly liked that the work combined primary data from the author and data collected from a literature search (DS2).

Response: This is awesome! Thank you!

  1. Generically about all figures that are Hierarchical clustering plots (e.g., Figures 2C & D). These plots must include notes and legends to help readers understand the conclusions.

Response: For better visualization, we have included all the notes and legends in the corresponding Supplemental Figures.

  1. Line 306 and 310. There are multiple places where it says "Error! Reference source not found".

Response: We have corrected these errors.  

Thank you!
